# Optimal Controller Design for Ultra-Precision Fast-Actuation Cutting Systems

**DOI:** 10.3390/mi13010033

**Published:** 2021-12-27

**Authors:** Fei Ding, Xichun Luo, Duo Li, Zheng Qiao, Bo Wang

**Affiliations:** 1Centre for Precision Engineering, Harbin Institute of Technology, Harbin 150001, China; fei.ding@hit.edu.cn (F.D.); liduo@hit.edu.cn (D.L.); qiaozheng_hit@126.com (Z.Q.); 2Centre for Precision Manufacturing, Department of DMEM, University of Strathclyde, Glasgow G1 1XJ, UK; xichun.luo@strath.ac.uk

**Keywords:** optimal control, ultra-precision machining, microstructure, controller tuning, fast tool servo

## Abstract

Fast-actuation cutting systems are in high demand for machining of freeform optical parts. Design of such motion systems requires good balance between structural hardware and controller design. However, the controller tuning process is mostly based on human experience, and it is not feasible to predict positioning performance during the design stage. In this paper, a deterministic controller design approach is adopted to preclude the uncertainty associated with controller tuning, which results in a control law minimizing positioning errors based on plant and disturbance models. Then, the influences of mechanical parameters such as mass, damping, and stiffness are revealed within the closed-loop framework. The positioning error was reduced from 1.19 nm RMS to 0.68 nm RMS with the new controller. Under the measured disturbance conditions, the optimal bearing stiffness and damping coefficient are $1.1\times 10^{5}\;N/m$ and $237.7\;N/\left(m\cdot s^{-1}\right)$, respectively. We also found that greater moving inertia helps to reduce all disturbances at high frequencies, in agreement with the positioning experiments. A quantitative understanding of how plant structural parameters affect positioning stability is thus shown in this paper. This is helpful for the understanding of how to reduce error sources from the design point of view.

## 1. Introduction

Ultra-precision freeform surfaces are widely used in advanced imaging and illumination systems, high-power beam-shaping applications, and other high-end scientific instruments [1]; they give the designers greater ability to cope with the performance limitations commonly encountered in simple-shape designs. However, the stringent requirements for surface roughness and form accuracy of freeform components pose significant challenges for current machining techniques—especially in the optical and display market, where large surfaces with tens of thousands of micro-features need to be machined [2,3,4].

The machining of such microstructures requires ultra-precision fast-motion systems. Typically, PID control laws are used; however, the PID control algorithm has only four free parameters that can be tuned, while the real-world situation is much more complex. The control algorithm and gains are often selected based on human experience through a “trial and error” process. It is possible to optimize the controller gains given the mathematical model of the system. E. A. Padilla-Garcia proposed a concurrent multi-objective dynamic optimization method to optimize the selection of controllers and motors; the optimization objectives were the energy consumption, tracking error, and motor weight. The efficiency of the proposed methodology was validated by simulations of an industrial robot [5]. Alter et al. applied an H−∞ robust control algorithm for controlling a linear motion stage, and the resulting dynamic stiffness was improved by 27–46% compared to PD control [6,7]; they further developed a stiffness-enhancement control law for optimal control, and the dynamic stiffness was improved by around 100% [8]. Dumanli utilized a linear–quadratic regulator (LQR) to achieve optimal placement of controller poles and zeros with acceleration feedback; he applied this algorithm in control of a ball screw feed drive, and achieved active damping with higher bandwidth [9]. Previous studies have mostly focused on the enhancement of dynamic stiffness. For precision fast-actuation cutting systems, the bandwidth and the positioning error are also very important. Wei-Wei Huang et al. developed a novel robust dual-loop control scheme with a Kalman-filter-based extended state observer and H∞ control for nano-positioning stages to implement high-bandwidth tracking operations [10]; they applied the control scheme on a piezo-driven stage, and the positioning bandwidth was improved from 3.6 kHz to 5.52 kHz. However, the positioning noise at this bandwidth is 20 nm, which is not sufficient for ultra-precision cutting systems. The positioning noise is mostly caused by environmental vibrations and the noises in the electronics. Feinan Zhu developed an improved reset control strategy to control the positioning of the read head in HDD; in his model, he included the external disturbances, and successfully reduced the tracking error in a finite time [11]. Parameter uncertainty can also be treated as a kind of disturbance. F. Mendoza-Mondragón proposed a two-degrees-of-freedom controller for robust speed regulation in permanent-magnet synchronous motors (PMSMs) [12]; the experimental results showed that better robust and disturbance rejection was achieved compared with traditional PI control. Chunhong Zheng proposed a simple but effective nonlinear proportional–derivative (PD) control strategy for faster, high-precision positioning [13]. It can be seen that the performance of positioning systems can be improved by optimizing the controller, but it is still difficult to predict the minimum tracking error before the hardware is built. Another issue with the design of fast positioning systems is that the tracking bandwidth and following error are greatly affected by the structural parameters, such as moving mass or damping. Li Zelong used multi-objective optimization and finite element simulation to design a flexure-hinge servo turret with a high natural frequency for fast tool servo applications [14]. It has been proven that the plant moving mass affects the minimum achievable positioning error [15]. Therefore, a system model that reveals the influences of structural parameters is necessary in order to achieve a quantitative understanding of how to reduce error from the design point of view.

In this paper, the uncertainty associated with controller tuning is precluded by adopting an H2 optimal control algorithm, which results in a control law minimizing positioning errors based on the plant model and measured disturbances. The minimum positioning errors are predicted with different structural parameters. A deterministic model to optimize the structural parameters to be minimized following error is proposed for the first time. Then, the influence of each structural parameter is analyzed. The results of our analysis reveal the optimal structural parameters and provide guidance on improving the dynamic performance of the tool positioning system. The control effects of the optimal controller and the PID controller are compared through a series of positioning tests.

## 2. Materials and Methods

In this section, we describe an optimal control strategy that was used to control a fast positioning system. A model to predict the static following errors was proposed based on the optimized controller. The results were used to study the influences of different structural parameters on positioning stability.

### 2.1. Experimental Setup

A custom-built fast tool servo cutting device for freeform turning is shown in Figure 1a. This configuration is based on a flat Lorentz actuator. Short-stroke high-frequency motions are achieved in the Z direction with flexure guidance. The motion along the X direction is driven by a linear motor and guided by a ball-bearing linear slide. A metrology straight edge is used as the position reference, and a capacitive displacement sensor is used to measure against it. The diamond cutter is fixed in the same line as the displacement sensor and the motor center. In this way, the force passes through the gravity center and the Abbey principle is obeyed, which is very important in reducing machine tool errors [16,17]. The detailed assembly design of the motor and the bearing structure is shown in Figure 1b.

### 2.2. Experimental Determination of the System Model

An accurate mathematical model of the mechanical and electrical systems was established prior to controller design. The lumped-parameter model of the mechanical system is established as shown in Figure 2. The *m*1, *k*1, *c*1 block represents the tool tip mounting flexibility, which reflects the dynamic performance of the tool holder and the coil support. *m*2 is the mass of the movable body, including the coil assembly and sensor. *k*2 and *c*2 are the stiffness and damping of the flexure bearing, respectively. *m*3 represents the mass of the X carriage. *m*4 is any flexible mass that will disconnect from *m*2 at high frequency. *k*5 and *c*5 are the stiffness and damping of the motor coil with respect to magnets, respectively.

Sweep sinusoidal signals are selected to test the response of the built positioning system. Sweep sinusoidal commands are sent from the D/A convertor, and the response of the open-loop system is measured by the capacitive displacement sensor. The lumped-parameter model of the system is known, and the mathematical model is used as the grey-box model. The model parameters are then estimated by fitting the grey-box model and the experimental data. The estimated parameters are listed in Table 1.

According to [18], the closed-loop control system can be represented by a transfer matrix G and a controller K, as shown in Figure 3. Disturbances are modelled as input w, while output performance to be evaluated is modelled as z. The controller senses the output y of the plant and then generates a control signal u to the plant. The column number of inputs w represents the number of disturbances. The transfer matrix G can be partitioned into four submatrices. Submatrix A represents the characteristic matrix of the plant in state-space denotation. The B1 block is the input matrix for all of the disturbances, while the last column (B2) corresponds to the control input u. The block C1 is the output matrix for the errors to be minimized, and the last row (C2) corresponds to the output measurement.

The H2-norm of a SISO system with transfer function J(s) is defined as [18]:(1)$$∥J{∥}_{2}={\left(\frac{1}{2\pi }\int_{-\infty }^{\infty }{\left|T\left(j\omega \right)\right|}^{2}d\omega \right)}^{1/2}$$

For a multivariable system with a transfer function matrix of$\;J\left(s\right)=\left[j_{mn}\right]$, the definition can be generalized to:(2)$$∥J{∥}_{2}={\left(\underset{mn}{\sum }∥j_{mn}∥{}_{2}^{2}\right)}^{1/2}={\left(\frac{1}{2\pi }\int_{-\infty }^{\infty }tr\left[T{\left(-j\omega \right)}^{T}T\left(j\omega \right)\right]d\omega \right)}^{1/2}$$

The matrix J(s) is the cost function, which is to be minimized in the optimization process. The selection of the cost function depends on the application requirements. In this case, the positioning error is to be minimized; thus, the transfer function matrix formed by the transfer function of each disturbance source to the tool position was selected as the cost.

The controller output u is also included in the cost function to be constrained, because there are hardware limits on the maximum controller output. The optimal control calculation is equivalent to solving a Riccati equation [19], and finally a controller transfer matrix K is calculated.

The input disturbances w in Figure 3 usually have colored spectrum characteristics, so the input can be modelled as a white noise going through a particular weighting filter. The transfer functions of the filters are then integrated into the plant model, and an augmented transfer matrix G is formed, as shown in Figure 4. $W_{e}$ and$\;W_{u}$ are the weighting filters for the positioning error and control output in the cost function, respectively. $W_{1}$,$\;W_{2}$, and $W_{3}$ are the weighting filters for the disturbances.

### 2.3. Modelling of Disturbances and Weighting Functions

The weighting function for the following errors $W_{e}$ controls the shape of the closed-loop sensitivity function. Since for a closed-loop system, the sensitivity at a high frequency range is always close to unity, $W_{e}$ is mainly for controlling the low frequency range sensitivity shape. According to [18], the weighting function can be expressed as:(3)$$W_{e}=\frac{s/M_{s}+\omega_{b}}{s+\omega_{b}\varepsilon }$$
where $M_{s}$ limits the peak response near the crossover frequency; $\omega_{b}$ is the intended closed-loop bandwidth; and *ε* is introduced to make the weighting function strictly proper, and its value should be selected according to the allowable static state following error under cutting force, namely, the static stiffness. In this analysis, $M_{s}$ is set to 1.2721 for critical damping, and ε is set to 1 × 10^−7^.

The weighting function for the output of the controller $W_{u}$ controls how much output will be commanded to achieve the desired performance. At frequency ranges higher than intended bandwidth, $W_{u}$ is used to limit the control output by adding a pole; thus, the response falls fast at high frequencies, in order to suppress sensor noise. The weighting function can be expressed as:(4)$$W_{u}=\frac{s+\omega_{bu}/M_{u}}{\varepsilon_{1}s+\omega_{bu}}$$
where $M_{u}$ and $\omega_{bu}$ limit the control output, and $\varepsilon_{1}$ is introduced to make the weighting function strictly proper. $M_{u}$ and $\omega_{bu}$ are set as large numbers (1 × 10^8^) to indicate that the motor power is enough for static positioning.

The disturbances are measured separately at each error source using a data acquisition board and a capacitive sensor. The disturbances are assumed to be stationary stochastic processes.

As shown in Figure 5a, the capacitive sensor noise is modelled as independent band-limited white noise. The weighting filters are valued as the square root of the signal’s average PSD value. The weighting filter for sensor noise is modelled as $W_{1}=1.36\times 10^{-6}\;\left(constant\right)$.

The current loop noise shown in Figure 5b is modelled with large amplitudes at the low frequencies, and the peak at 7748 Hz is modelled by a poorly damped second-order peak filter. The weighting filter can be modelled as:(5)$$W_{2}=1.64\times 10^{-7}\times \frac{s^{2}+0.1\omega s+\omega^{2}}{s^{2}+0.005\omega s\times \omega^{2}}\times \frac{s-200\pi }{s}$$
where ω=2π×7748. Then, the environmental disturbance vibrations are chosen as follows:(6)$$W_{3}=\frac{2.83\times 10^{-9}\left(s-2000\pi \right)}{s}$$

### 2.4. Modelling of Following Errors

With the system model, the frequency response functions from each disturbance input to the tool position $FRF_{i}\left(\upsilon \right)$ can be obtained. The error power contribution $PSD_{i}$ from each disturbance to the final position can be calculated as shown in Equation (7):(7)$$PSD_{i}\left(\upsilon \right)=P_{i}\left(\upsilon \right)\times \left|FRF_{i}\left(\upsilon \right)\right|{}^{2}$$
where *i* indicates the disturbance source number (from 1 to 3; *P*_1_, *P*_2_, and *P*_3_ are the PSD of each error source, and *υ* is the frequency). Since these disturbances are assumed to be mutually uncorrelated, their powers can be combined to reflect the total error power [20]. The synthesized tool position’s PSD is:(8)$$PSD_{Fol}\left(\upsilon \right)=P_{1}\left(\upsilon \right)\times \left|1-FRF_{1}\left(\upsilon \right)\right|{}^{2}+PSD_{2}\left(\upsilon \right)+PSD_{3}\left(\upsilon \right)$$

In order to estimate the time-domain error magnitude from the *PSD* values, the cumulative amplitude spectrum (*CAS*) function is derived. $CAS_{i}\left(\upsilon \right)$ is the square root of the integrated $PSD_{i}\left(f\right)$, from 0 Hz to υ Hz.
(9)$$CAS_{i}\left(\upsilon \right)=\sqrt{\int_{0}^{\upsilon }PSD_{i}\left(f\right)df}$$

The following error can be calculated as follows:(10)$$Error_{rms}=CAS_{i}\left(\upsilon_{Nyquist}\right)$$
where $\upsilon_{Nyquist}$ is the Nyquist frequency—namely, the frequency span.

## 3. Results

### 3.1. Closed-Loop Response with Optimal Control

Using the above model, the solved optimal controller that minimized the following errors was a 27 × 27 matrix in the state-space form with 27 state variables. The open-loop and closed-loop transfer functions of the modelled system with the calculated optimal controller are shown in Figure 6a. The red line K is the frequency response of the calculated controller. The dashed black line is the modelled structure transfer function. The blue line L is the estimated open-loop transfer function, and the black solid line T is the estimated closed-loop transfer function. The dashed magenta line D is the estimated disturbance rejection function.

In comparison, the same crossover frequency was achieved with a PID algorithm, and the calculated controller functions are shown in Figure 6b. There exists a structural resonant point at frequency of 1645 Hz, which can cause troubles when the PID gains are further increased. This peak is successfully compensated in the optimal controller, because the controller has more control of degrees of freedom. The low-frequency control actions are also different in that the optimal controller is fully determined by the disturbance strengths, while the PID controller is calculated according to the phase margin set by the user.

The measured following errors with the optimal controller are shown in Figure 7. The position bandwidth (−3 dB) is found to be around 1.1 kHz. The RMS value is 0.68 nm and the peak-to-valley value is 5.38 nm. The calculated positioning error is 0.23 nm RMS by the model prediction, and the peak-to-valley value should be around 1.4–2.3 nm.

In comparison, the PID controller is tuned with the same sampling rate and a similar bandwidth (1.5 kHz), and the following errors are measured as shown in Figure 8. Because the sampling rate is kept the same, the measuring error contribution from the feedback sensor should be the same. The following errors are larger when the bandwidth is increased, with an RMS value of 1.19 nm and a peak-to-valley value of 7.65 nm with PID control. These results show that the optimal controller indeed helps to achieve better positioning stability.

The FFT spectra of the two error signals are shown in Figure 9. It can be seen that the error spectrum is more evenly distributed when the optimal controller is utilized. The high-frequency noise is higher when the PID controller is used.

### 3.2. Study on the Influences of Structural Parameters on Positioning Following Error

From the control point of view, the controller has theoretically done its best to suppress outside disturbances. If the disturbances cannot be reduced from the roots, it is worthwhile to study how to reduce the system response to the disturbances by changing the structural parameters, such as mass, damping, etc. Several selected parameters were studied for their influences on the positioning following errors based on the closed-loop model, including the total mass of the moving part, flexure stiffness, damping, and motor force constant. The effects of changing plant parameters need not be linear. Thus, the current system design is used as the “operating point”. The structural parameters are changed, and the resultant following RMS errors are used to compare the sensitivities. This helps to figure out the most effective way to optimize the performance.

#### 3.2.1. Influence of Moving Mass

When the total moving mass $m_{1}+m_{2}$ is doubled based on the current configuration, all high-frequency noises are reduced, as shown in Figure 10a. The current noise transfer functions are notably reduced at high frequencies. Base vibration errors are also reduced at high frequency. The side effect of increasing moving mass is that more force is needed to achieve the same acceleration. This means bigger motors will be used and, thus, more heat will be generated. The CAS plot in Figure 10b also shows that the positioning errors are lower.

The minimum achievable positioning errors with different moving masses are shown in Figure 11a. It can be seen that the errors decrease with larger mass, as does the bandwidth (−3 dB). The minimum achievable error is plotted in Figure 11b. The RMS error also decreases with increased moving mass, but it is larger than that in Figure 10a because it deviates from the optimal bandwidth.

#### 3.2.2. Influence of Flexure Bearing Stiffness and Damping

The flexure bearing in the designed system is the only path that external vibrations can travel to the tool. Therefore, the stiffness $k_{2}$ and $c_{2}$ affect the degree to which environmental vibrations will be transferred to the tool tip. Meanwhile, they also affect the rejection ability of force disturbances; this can be seen in Figure 12a and Figure 13a. When the flexure stiffness $k_{2}$ increases, the transfer function from base vibrations is raised at low frequency. Meanwhile, when the damping $c_{2}$ increases, more high-frequency base vibration is transmitted to the tool. However, the errors caused by the current-stage noise are reduced in both cases, while the optimal closed-loop bandwidth is also decreased slightly.

The CAS functions are plotted in Figure 12b and Figure 13b. The total following error is decreased because the contribution of the base vibration is so small. This is not always true; when the stiffness or damping is increased to such a level that the base vibration contribution is comparable to the reduction in the current noise contribution, the total error will be increased.

There exists an optimal pair of flexure stiffness $k_{2}$ and damping $c_{2}$ coefficients under this disturbance situation, which minimizes the following errors, as can be seen in Figure 14. In this analysis, the stiffness $k_{2}$ and damping $c_{2}$ are adjusted within a large range, and the total RMS following error is plotted. As the stiffness and damping coefficients are increased, the following error first deceases and then starts to rise. The minimum following error of RMS (0.96 nm) is achieved when $k_{2}$ is equal to $1.1\times 10^{5}\;N/m$ and $c_{2}$ is equal to$\;237.7\;N/\left(m\cdot s^{-1}\right)$. Apparently, this optimal stiffness and damping value is dependent on the relative strength of the disturbances.

## 4. Conclusions

In this paper, optimal control was achieved for a fast-actuating motion system. The influences of mechanical parameters such as mass, damping, and stiffness were investigated. The following conclusions can be drawn:

The positioning error was reduced from 1.19 nm RMS to 0.68 nm RMS with the new controller, showing the benefits of a deterministic controller design approach;Under the given disturbances, there exist optimal bearing stiffness and damping coefficients that result in minimal following errors. The optimal bearing stiffness and damping coefficients are $1.1\times 10^{5}\;N/m$ and $237.7\;N/\left(m\cdot s^{-1}\right)$, respectively;It was found that increasing moving mass helps to reduce following errors, but the optimal bandwidth will be smaller.

## 5. Future Work

The current analysis studied the positioning stability of tools when they hold their position, which is applicable for cutting of flat surfaces. When the tool starts to follow the high-frequency profiles, there are other disturbances, such as the inertia forces and cutting forces. Therefore, more detailed modelling of such disturbances can be added in future works.

## Figures and Tables

**Figure 1 micromachines-13-00033-f001:**
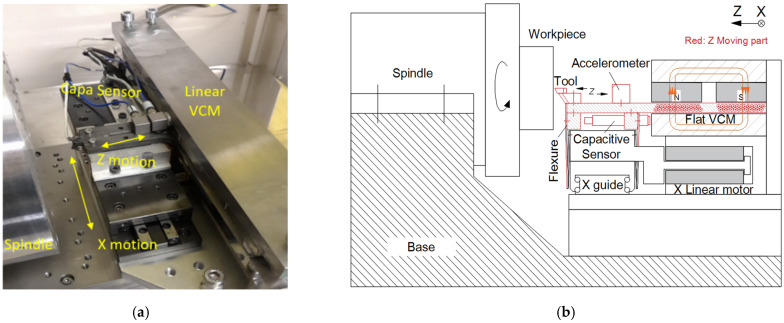
(**a**) Photo and (**b**) structural diagram of the fast-actuation cutting system.

**Figure 2 micromachines-13-00033-f002:**
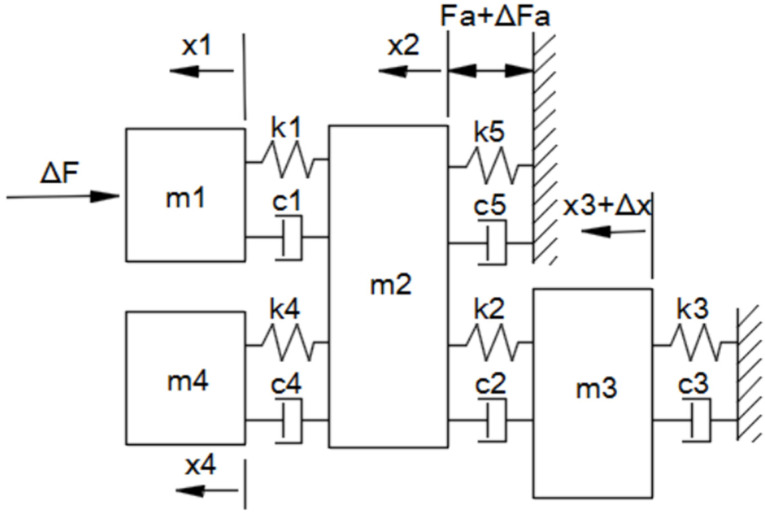
Lumped-parameter model of the positioning system.

**Figure 3 micromachines-13-00033-f003:**
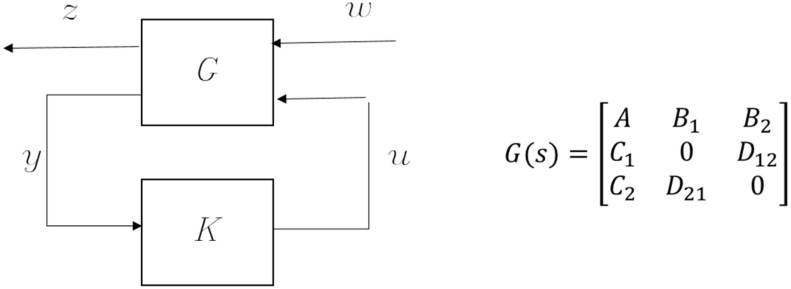
Standard representation of the H2 optimal control problem.

**Figure 4 micromachines-13-00033-f004:**
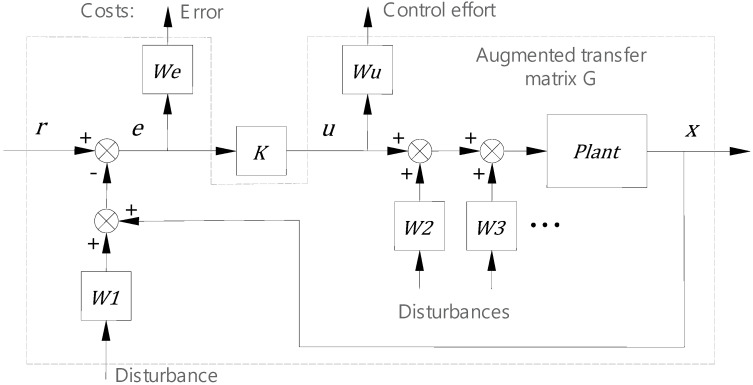
Augmented plant configuration for optimal controller design.

**Figure 5 micromachines-13-00033-f005:**
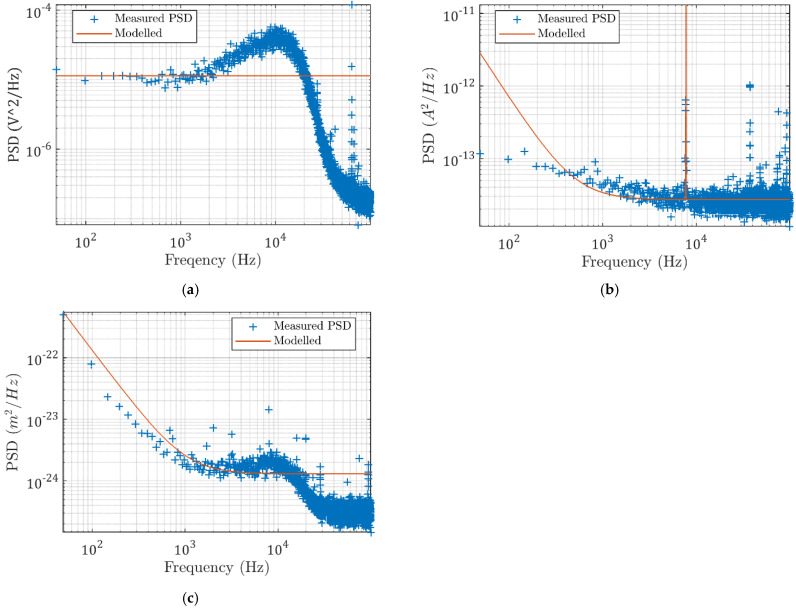
Measured (**a**) sensor noise (**b**), current loop noise, and (**c**) environmental vibrations and the modelled PSD functions.

**Figure 6 micromachines-13-00033-f006:**
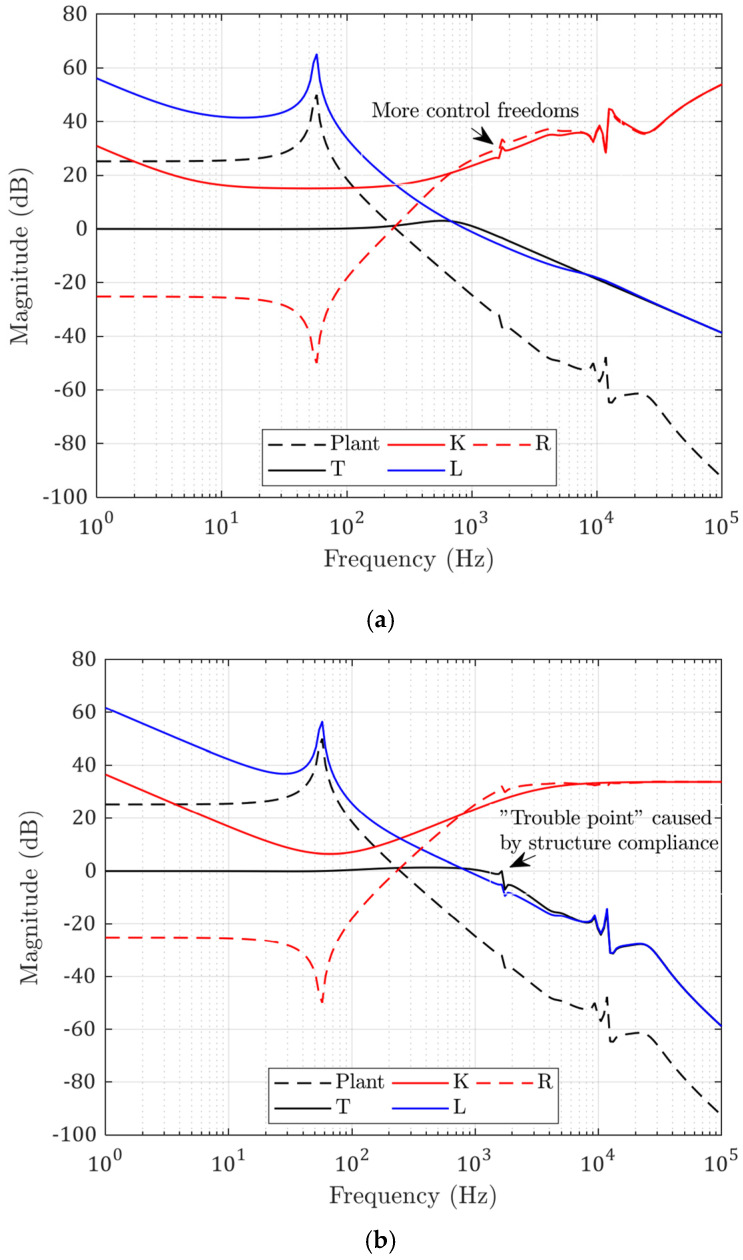
(**a**) Calculated optimal controller and transfer functions; (**b**) controller and transfer functions with PID control.

**Figure 7 micromachines-13-00033-f007:**
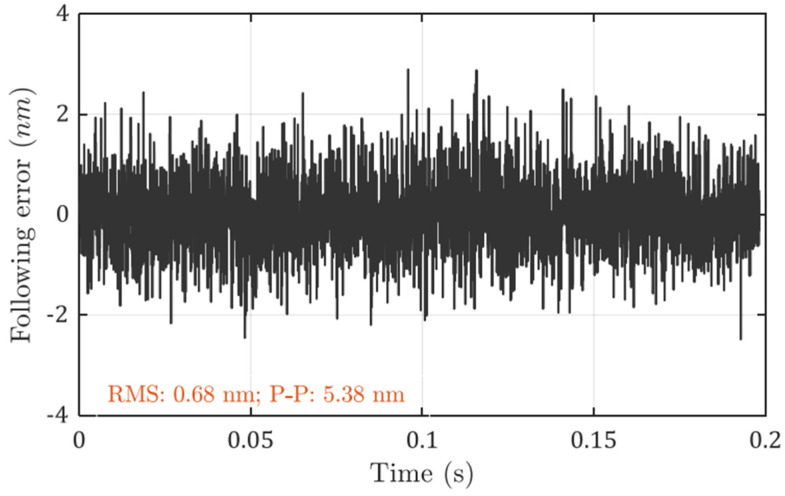
Measured following error with optimal controller at a bandwidth of 1.1 kHz.

**Figure 8 micromachines-13-00033-f008:**
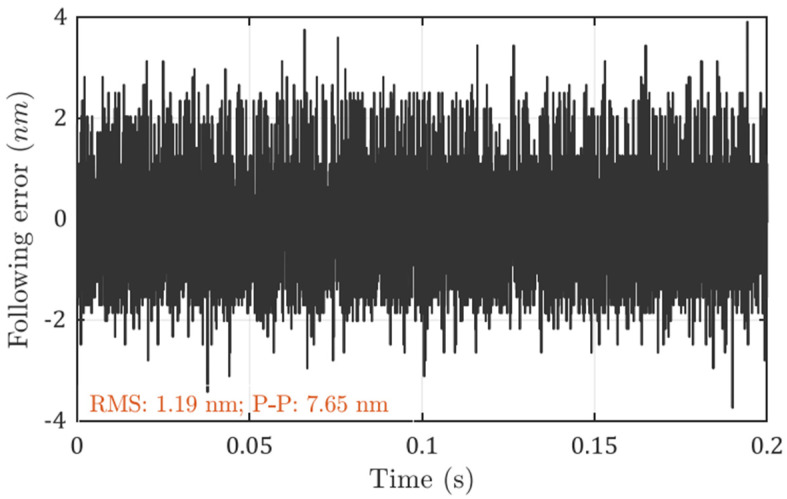
Measured following error with PID controller.

**Figure 9 micromachines-13-00033-f009:**
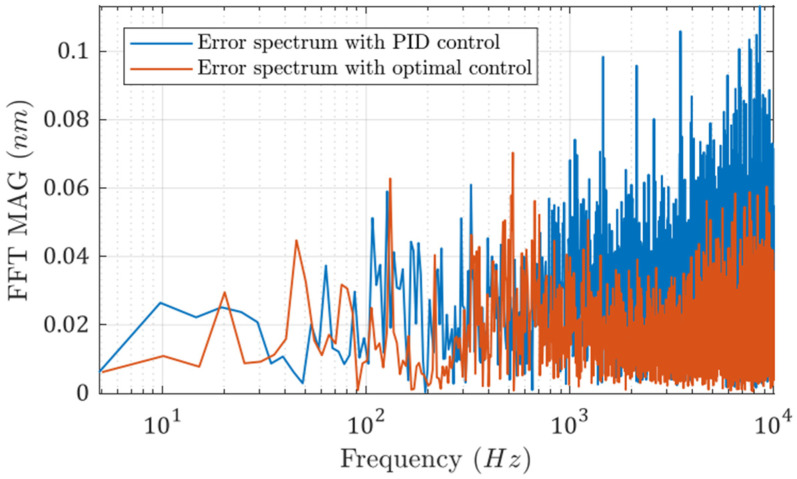
FFT spectra of the two error signals with different control algorithms.

**Figure 10 micromachines-13-00033-f010:**
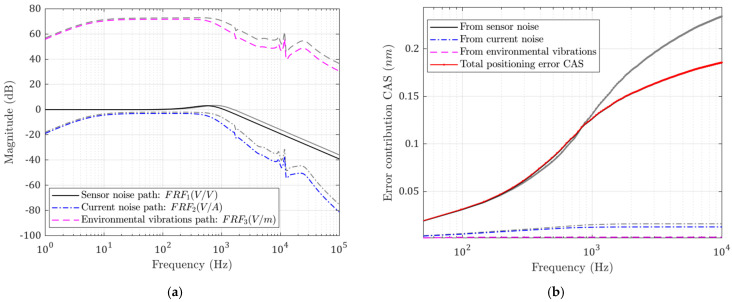
(**a**) Noise transfer functions; (**b**) CAS plots for each error source. Performance with increased moving mass (grey: before; colored: after).

**Figure 11 micromachines-13-00033-f011:**
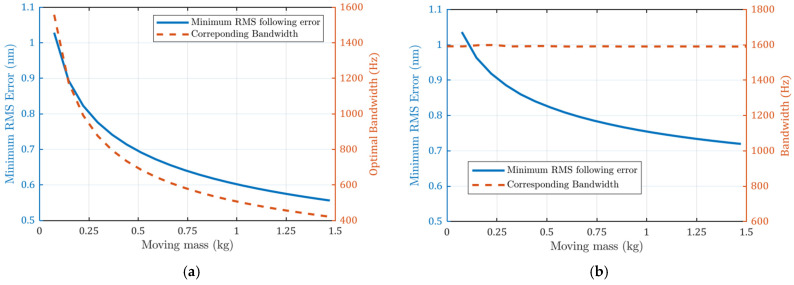
(**a**) Errors with variable optimal bandwidth; (**b**) errors with constant bandwidth. Minimum achievable positioning error decreases with larger moving masses.

**Figure 12 micromachines-13-00033-f012:**
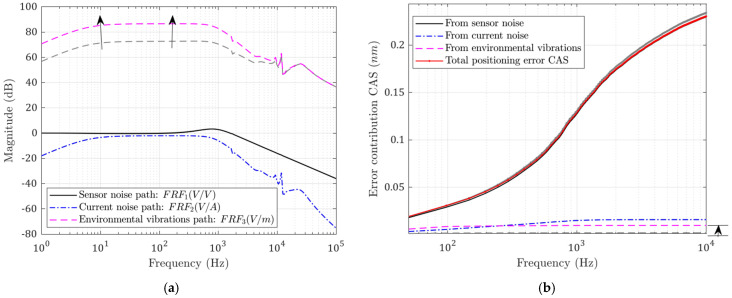
(**a**) Noise transfer functions; (**b**) CAS plots for each error source. Performance with increased flexure stiffness $k_{2}$ (grey: before; colored: after).

**Figure 13 micromachines-13-00033-f013:**
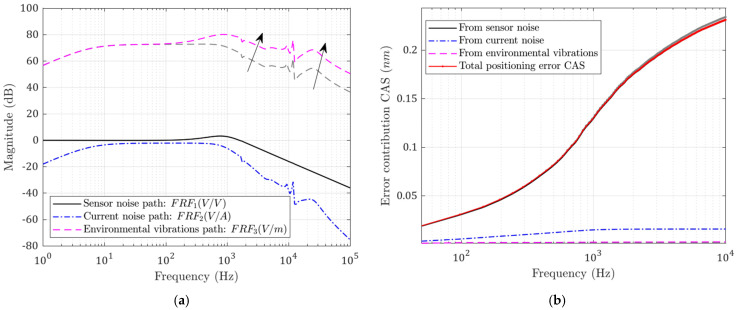
(**a**) Noise transfer functions; (**b**) CAS plots for each error source. Performance with increased flexure damping $c_{2}$ (grey: before; colored: after).

**Figure 14 micromachines-13-00033-f014:**
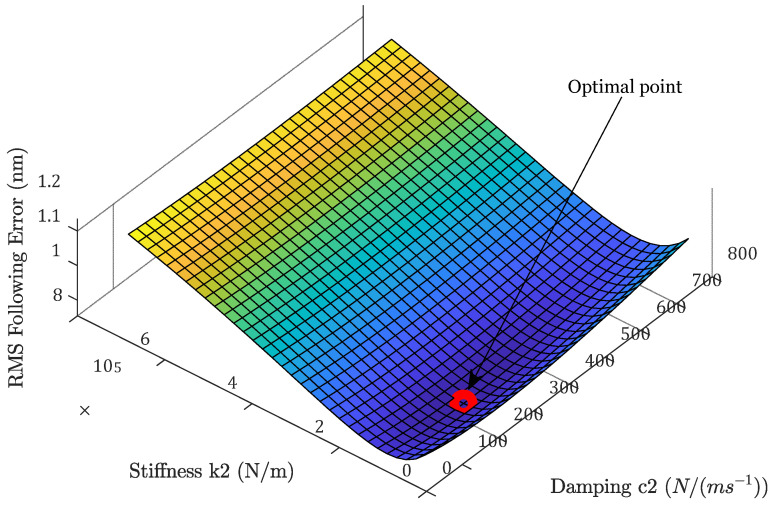
Optimal bearing stiffness and damping under the current disturbance situations.

**Table 1 micromachines-13-00033-t001:** Identified parameters of the developed positioning system.

Parameters	Values
$m_{1}+m_{2}$	0.074 Kg
$k_{2}$	21,965 N/m
$c_{2}$	2.28 N/(m/s)
$m_{4}$	0.008 Kg
$k_{4}$	56,285 N/m
$c_{4}$	6.18 N/(m/s)

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
