# Peer review of "Optimal Controller Design for Ultra-Precision Fast-Actuation Cutting Systems"

_micromachines, 2021, doi:10.3390/mi13010033_

Round 1
Reviewer 1 Report
The subject of the paper corresponds to the topic approached by Micromachines. The issue is interesting and current.
The paper may be published after several corrections have been made.
I consider that the presentation of the current state of research is not enough. Information on the optimal H2 control algorithm is required. The bibliography could be improved.
The authors can present information on the numerical values ​​of the measured parameters. The experimental plan and the results on the basis of which the diagrams were drawn could be presented. Information is needed on how to obtain the optimal point. It was obtained only graphically or and mathematically. I believe that clearer conclusions could be drawn on the results of the research (partially presented in the subchapter "Discussions").
Authors should pay attention to formal issues. For example, there is not enough information on Figure 2, right side. Also, the references in the text on figures 3 - 13 are not completed (lines, 106, 136, 147, 156, 160, 169, 206, 207, 217, 222, 234).
Author Response
- Responses to Reviewer #1:
Point #1: I consider that the presentation of the current state of research is not enough.
Response #1: Thanks for your comments. I have extended the range of the current status of research in the introduction section.
Point #2: Information on the optimal H2 control algorithm is required.
Response #2: Thanks for your comments. I have added a reference on the optimal H2 control theory, and I also put more details on the selection of weighting functions.
Point #3: The bibliography could be improved.
Response #3: Thanks for your comments. I have added more recent literatures such as [4-5], [10-14], [19,20]
Point #4: The authors can present information on the numerical values ​​of the measured parameters.
Response #4: Thanks for your comments. The meaning of the measured parameters is described in section 2.2
Point #5: The experimental plan and the results on the basis of which the diagrams were drawn could be presented.
Response #5: Thanks for your comments. I have added a paragraph about the global idea of the experimental plan at the beginning of section 2. The references to the figure and equations has been added.
Point #6: Information is needed on how to obtain the optimal point. It was obtained only graphically or and mathematically. I believe that clearer conclusions could be drawn on the results of the research (partially presented in the subchapter "Discussions").
Response #6: Thanks for your comments. Since the approach to calculate following errors from disturbances is based on numerical simulation and cannot find an analytical solution. The optimal point is found through calculations at a series of discrete points. The Discussion section has been revised.
Point #7: Authors should pay attention to formal issues. For example, there is not enough information on Figure 2, right side. Also, the references in the text on figures 3 - 13 are not completed (lines, 106, 136, 147, 156, 160, 169, 206, 207, 217, 222, 234).
Response #7: Thanks for your comments. I have corrected the mistakes on Figure 2. The references to each figure in the text has been highlighted.
Reviewer 2 Report
The authors present the article entitled “Optimal controller design for ultra-precision fast actuation cutting system”. However, it is not possible to extend my recommendation for publication in its current form according to the next concerns.
The manuscript in its current form presents older literature
Introduction section: The novelty of the work is not clear. What is the main contribution of the work?
Materials and methods section needs a hard improvement. The theoretical background is not clear. References are not mentioned to support the methodology, and equations are not numbered.
Line 129: Explain in detail how the weights can be tuned.
Discussion section must improve: This section must show the findings and their implications in the broadest context possible.
The drawbacks of PD, PID and other controllers can be discussed between lines 32-40 by considering: Robust speed control of permanent magnet synchronous motors using two-degrees-of-freedom control; Concurrent optimization for selection and control of ac servomotors on the powertrain of industrial robots; Fpga-based architecture for sensing power consumption on parabolic and trapezoidal motion profiles.
Vectorize the figures to see the details.
Include a quantitative value in the abstract to highlight the findings.
Include a table which compares the findings of the work vs the already reported in the stat of the art.
Include quantitative results in the last section.
There are lots of already fresh papers well related in Micromachines to be taken into account as references as Characterization of Surface Topography Variation in the Ultra-Precision Tool Servo-Based Diamond Cutting of 3D Microstructured Surfaces
Comprehensive Design Method of a High-Frequency-Response Fast Tool Servo System Based on a Full-Frequency Error Control Algorithm
Conclusion section: Even though this section is not mandatory, I suggest adding this section according to the main objective of the manuscript. Also, future works must be mentioned.
Author Response
- Responses to Reviewer #2:
Point #1: The manuscript in its current form presents older literature
Response #1: Thanks for your comments. I have added more recent literatures such as [4-5], [10-14], [19,20]
Point #2: Introduction section: The novelty of the work is not clear. What is the main contribution of the work?
Response #2: Thanks for your comments. Applying optimal control theory in motion system is not rare. However, they have mostly focused on improving the stiffness or robustness. In this paper, I applied the theory on reducing the following errors, which directly affect the surface roughness of the machined parts. Furthermore, with the ability of predicting following errors from noise models, I have revealed how the system structural parameters affects the achievable best performance. This has not been reported in other literatures.
Point #3: Materials and methods section needs a hard improvement. The theoretical background is not clear. References are not mentioned to support the methodology, and equations are not numbered.
Response #3: Thanks for your comments. I have added a reference on the optimal H2 control theory, and I also put more details on the selection of weighting functions. Equations have been numbered.
Point #4: Line 129: Explain in detail how the weights can be tuned.
Response #4: Thanks for your comments. I have added a separate section on how the weighting functions are obtained.
Point #5: Discussion section must improve: This section must show the findings and their implications in the broadest context possible.
Response #5: Thanks for your comments. I have revised the Discussion section.
Point #6: The drawbacks of PD, PID and other controllers can be discussed between lines 32-40 by considering: Robust speed control of permanent magnet synchronous motors using two-degrees-of-freedom control; Concurrent optimization for selection and control of ac servomotors on the powertrain of industrial robots; Fpga-based architecture for sensing power consumption on parabolic and trapezoidal motion profiles.
Response #6: Thanks for your comments. I have included the related reference in the introduction part.
Point #7: Vectorize the figures to see the details.
Response #7: Thanks for your comments. The figures have been replaced by higher quality ones
Point #8: Include a quantitative value in the abstract to highlight the findings.
Response #8: Thanks for your comments. I have revised the abstract accordingly and included quantitative values of the research.
Point #9: Include a table which compares the findings of the work vs the already reported in the stat of the art.
Response #9: Thanks for your comments. Since the similar works by other researchers are focused on different aspects of positioning system. Some discussed robustness or stiffness of optimal control approaches, rather than positioning following errors. Therefore, it is a little bit difficult to summarize into a table. However, I have included numerical values of following errors where applicable such as ref [10]
Point #10: Include quantitative results in the last section.
Response #10: Thanks for your comments. I have revised the Discussion section and change it to point by point conclusions.
Point #11: There are lots of already fresh papers well related in Micromachines to be taken into account as references as Characterization of Surface Topography Variation in the Ultra-Precision Tool Servo-Based Diamond Cutting of 3D Microstructured Surfaces; Comprehensive Design Method of a High-Frequency-Response Fast Tool Servo System Based on a Full-Frequency Error Control Algorithm
Response #11: Thanks for your comments. I have included the relevant reference at [4] [14].
Point #12: Conclusion section: Even though this section is not mandatory, I suggest adding this section according to the main objective of the manuscript. Also, future works must be mentioned.
Response #12: Thanks for your comments. I have combined the discussion section and the conclusion section. Future works are mentioned as well.
Round 2
Reviewer 1 Report
The authors made improvements to the original manuscript. I consider that the paper can be published.
Reviewer 2 Report
The manuscript is adequate for publishing.